# COVID-19 and Attendance Demand for Professional Sport in Japan: A Multilevel Analysis of Repeated Cross-Sectional National Data during the Pandemic

**DOI:** 10.3390/ijerph19095318

**Published:** 2022-04-27

**Authors:** Hiroaki Funahashi, Shintaro Sato, Takuya Furukawa

**Affiliations:** 1School of Health and Sport Sciences, Chukyo University, 101 Tokodachi, Kaizu-cho, Toyota 470-0393, Japan; 2Faculty of Sport Sciences, Waseda University, 3-4-1 Higashifushimi Nishi-Tokyo, Tokyo 202-0021, Japan; satoshintaro@aoni.waseda.jp; 3Department of Sports Management, Faculty of Management, Osaka Seikei University, 3-10-62 Aikawa Higashiyodogawa-ku, Osaka 533-0007, Japan; furukawa-t@g.osaka-seikei.ac.jp

**Keywords:** COVID-19, spectator demand, professional sports, multilevel logistic model

## Abstract

In the current investigation, we assess the effect of COVID-19 on intention-based spectator demand for professional sports in Japan captured by eight, monthly repeated cross-sectional national surveys from May to December 2020 (*n* = 20,121). We regress spectator demand on individual (e.g., gender), prefecture-wave (e.g., COVID-19 infection status), and prefecture-level factors (i.e., with or without quality professional teams). The results of multilevel logistic regression demonstrate that individual (i.e., male, younger, full-time employment, and with children status) and prefecture-level team factors (i.e., with teams) were associated with intention-based spectator demand. Nevertheless, COVID-19-related factors were found to be unrelated to spectator demand. The findings imply that sports fans are likely to return to the stadium once behavioral restrictions are lifted. The current research provided further evidence that individual factors and team quality serve as influential antecedents of spectator demand in the context of the COVID-19 epidemic.

## 1. Introduction

Given the recent global outbreak of coronavirus disease (COVID-19), caused by SARS-CoV-2, the role of social distancing and economic activity in the transmission of viruses is of growing research interest. Recent evidence on the latter reports a positive impact of exogenous large gatherings during the pandemic on local COVID-19 transmission, including university students returning from spring break [1], in-person voting [2], professional/college team sport, and motorcycle events [3,4,5,6]. The literature also shows that the exogenous relocation [7] and success [8] of sports franchises contributes to the spread of influenza. In short, the previous studies consistently suggest that mass gatherings at sporting events cause contagious externalities.

Against this backdrop, the question arises as to whether the public health threat keeps sports fans away from stadiums. Gitter [9] examined stadium attendance demand for Mexican League Baseball games during the H1N1 pandemic, demonstrating that the spread of influenza-like illness negatively affects attendance. Reade and Singleton [10] analyzed the response behavior of football spectators to the COVID-19 outbreak in five European countries during the early 2020 pandemic, showing that there was not necessarily a negative relationship between the increased threat of COVID-19 in the country and spectator demand. Reade et al. [11] studied attendance demand for the Belarusian Premier League, a football league that kept playing despite the COVID-19 pandemic, finding that football fans reacted negatively to the threat of COVID-19 but a gradual habituation effect arose. Another study, with a slightly different focus, investigating the behavioral response of sport consumers to a polluted environment concluded that the consumption habits of football fans do not change despite the presence of air pollution [12]. In view of these mixed results, in order to elaborate on the impact of external shocks to health threats on spectator behavior, it would be useful to observe consumer’s psychological response, i.e., the preferences and intention of sport consumers as a precursor to attendance.

In this paper, we estimate the effect of the local prevalence of COVID-19 on intention-based spectator demand. We compiled a unique dataset combining data from eight repeated cross-sectional surveys on the spectator demand for professional sports among the Japanese public during pandemic, with COVID-19 infection status in the 47 prefectures (sub-national jurisdictions). Our study contributes to the literature examining the psychological or behavioral response of individuals in public health crises and, more specifically, to the spectator demand literature in terms of modeling multilevel factors (i.e., individual, prefecture-wave, and prefecture level) on micro-data on individual spectator intentions collected by repeated cross-sectional surveys of a representative national sample. It also addresses the call for research on stadium attendance demand in Asia, where little evidence exists [13].

The setting of the present study is the Japanese spectator sports market. In Japan, the first COVID-19 case was identified in January 2020 [14]. Subsequently, the government asked organizers of professional sport games to voluntarily self-restrain their events for a total of about four months starting in late February 2020. With the nationwide lifting of the declaration of a state of emergency (7 April–25 May), the government announced guidelines for the gradual relaxation of restrictions on the hosting of events [15]. Following the guidelines, professional sporting events have been phased in and relaxed to (1) no spectators; (2) up to 50% of the stadium capacity or 5000 people, whichever is less; and eventually (3) up to 50% of the capacity.

Amid the COVID-19 outbreak, Nippon Professional Baseball (NPB) and the J.League formed a COVID-19 task force to assess how to respond to the virus, inviting infectious disease experts as advisors. The joint steps taken by two of the biggest professional sports circuits attracted public attention, and the proposed COVID-19 measures were widely shared with other sports organizations, making a significant contribution to the Japanese spectator sports industry. To elaborate, the task force developed infection control guidelines and put in place measures to prevent infection among players and officials, stadium operations, response to outbreaks of the disease, and immigration control for foreign players. Spectators were also required to take precautions against COVID-19, including temperature checks and disinfection at the entrance to the stadium, seating with plenty of space between seats, the prohibition of the sale of alcoholic beverages and the prohibition of cheering. To date, there have been no reports of COVID-19 clusters at professional sporting event venues in Japan.

The aim of the present paper is to examine the effect of COVID-19 on intention-based spectator demand for professional sports in Japan. The remainder of the paper is organized as follows. Section 2 (Methods) describes the data and empirical strategy. Section 3 (Results) provides the estimation results. Section 4 (Discussion) elaborates on the findings, and conclusions are presented in Section 5.

## 2. Methods

### 2.1. Data Sources and Variables

Individual spectator demand data were collected from eight rounds of the Spectator Demand Survey (SDS), a joint research initiative conducted by the Research Institute of Sport Business Waseda University, the Research Institute of Sport Management Doshisha University, and the Japan Society of Sports Industry. The SDS is a public opinion survey designed in 2020 to help determine the timing of the resumption of sporting events that have had to be canceled or postponed due to the COVID-19 pandemic. It is an Internet-based repeated cross-sectional survey conducted once every two weeks (changed to once a month in the middle of the survey) starting from 15 May 2020, targeting approximately 2500 people aged 18 years or older nationwide. For the analysis of this study, data from eight one-month intervals from 29 May 2020 (round 1) to 11 December 2020 (round 8) were used (*n* = 20,121). Data are available to members of the Japan Society of Sports Industry.

The outcome variable, spectator demand, was based on the survey participant’s response to the question regarding stadium/arena attendance demand for games in three professional ball sports leagues in Japan, i.e., Nippon Professional Baseball (NPB), Japan Professional Football League (J.League), and Japan Professional Basketball League (B.League). Specifically, the question was, “Are you willing to watch sports in a stadium or arena in the future? Please select all the leagues in which you would like to watch a game.” The responses were used to generate individual dichotomous spectator demand data for each professional sport league (1 = *NPB/J.League/B.League demand* yes, 0 = otherwise).

Variables on three different levels—Level 1: individual, Level 2: prefecture-wave, and Level 3: prefecture—were used as independent variables (for a descriptive overview see Table 1). First, Level 1 independent variables were sociodemographic characteristics of respondents including gender, age, employment status, marital status, and number of children. All demographic variables were dummy coded as shown in Table 1 except for age, which was a continuous variable.

Second, the variables of interest, *Cases* and *Deaths*, entered the analysis on Level 2 and were operationalized as the number of newly confirmed COVID-19 cases/deaths in the week leading up to the day before the survey start date. Note that for these COVID-19 variables, we used group-mean centering at prefecture-level because we believed that relative differences in infection status within the same region would provide more-meaningful data than absolute numbers of cases or deaths. *Cases* was divided by 100 to obtain manageable numerical expressions. Information on the numbers of confirmed cases from COVID-19 and related deaths were obtained from NHK’s dedicated COVID-19 website [16]. The economic condition in each prefecture (*Economy*) was also added as a Level 2 economic control expected to influence demand for attendance at sporting events [17]. These economic data were proxied by the monthly economic diffusion index (DI) for each region as reported by Teikoku Databank (TDB) Economic Trend Research [18]. The TDB Economic Trend Research is a monthly business survey of more than 20,000 companies nationwide, primarily for the purpose of understanding the current state of the Japanese economy. Companies are asked to rate their business confidence on a seven-point scale from “very bad (0)” to “extremely good (6)”, and the DI is calculated by multiplying the score by the percentage of each response category. A DI over 50 means “good” economy and below 50 means “bad”.

Third, for prefecture-level (Level 3) controls, we introduced variables distinguishing between prefectures with and without professional sport teams as prefecture characteristic that remain constant over survey rounds. Considering the hierarchical structure of the leagues, dummy variables were created as follows: *NPB*, *J1 League*, *J2 League*, *J3 League*, *B1 League*, and *B2 League* (1 for prefectures with teams belonging to the relevant league, 0 otherwise).

Finally, we added wave dummies (wave 1 as reference) in order to control for time trends in public attendance demand for spectator sports. This was also meant to control for the impact of each league being in-season and changes in the government’s relaxation of attendance limits.

### 2.2. Multilevel Analysis

Data were analyzed using multilevel logistic models, treating respondents (at Level 1) as nested in prefecture-wave (at Level 2) and as nested in prefectures (at Level 3). The analyses were conducted in two steps. First, we analyzed the variation in spectator demand without using explanation variables to decompose the variance of the intercept into the variance components for each of the three levels. Such models are called intercept-only models or variance component models. The second step was to analyze a model with all the above explanatory variables fixed and observe how the three levels of variables affect the spectator demand (during a pandemic). The three-level logistic intercept-only model can be written as follows:(1)$$logit\left(Y_{ijk}\right)=\beta_{0}+e_{ijk}+u_{0jk}+v_{0k}$$
where *Y_ijk_* is a dichotomous variable indicating whether individual *i* in prefecture *k* at prefecture-wave *j* has an attendance demand for NPB, J.League, and B.League games or not. The intercept (*β*_0_) varies randomly across prefecture-waves and prefectures, which is indicated by the subscripts *j* and *k*. Furthermore, *e_ijk_* denotes the residuals at the individual-level. The intercept of the *j*th prefecture-wave represents the average intercept *β*_0_ plus a random deviation *u*_0*jk*_. Following the same principle, the intercept of the *k*th prefecture is the mean intercept *β*_0_ plus the random variation *v*_0*k*_. These random terms (i.e., *u*_0*jk*_ and *v*_0*k*_) and the residual errors (i.e., *e_ijk_*) are assumed to be mutually independent and normally distributed with mean zero and variance *σ*^2^. The specification of the model for the second stage, which includes explanatory variables, is described as follows:(2)$$logit\left(Y_{ijk}\right)=\beta_{0}+\beta_{1}X_{1ijk}+\beta_{2}X_{2jk}+\beta_{3}X_{3k}+\beta_{4}wave_{j}+e_{ijk}+u_{0jk}+v_{0k}$$

Individual-level variables are denoted as *X*_1*ijk*_, prefecture-wave-level variables introduced in Level 2 as *X*_2*jk*_, and prefecture-level time-invariant variables as *X*_3*k*_. Among them, our focus is on the impact of the variable that capture the intra-regional transmission of COVID-19, *Cases* and *Deaths*, encoded in *X*_2*jk*_. The model includes time dummies for each round (*wave_j_*) to rule out the possibility of time-specific effects that are common to all prefectures in the sample.

For each model, we computed the variance partition coefficient (VPC), which refers to the percentage of the total variance attributable to each level. The VPC at the prefecture level can be calculated as [*v*_0*k*_/(*v*_0*k*_ + *u*_0*jk*_ + 3.29)], the VPC at the prefecture-wave level as [*u*_0*jk*_/(*v*_0*k*_ + *u*_0*jk*_ + 3.29)], and the VPC at the individual level as [3.29/(*v*_0*k*_ + *u*_0*jk*_ + 3.29)] [19,20]. We display these values as a percentage of the total variance (VPC × 100).

## 3. Results

### 3.1. Descriptive Statistics of Sociodemographics

The pooled database from wave 1 to 8 contained 20,121 observations. The mean age was 50.1 (SD 16.9), 47.9% were male, 45.4% were in full-time employment, 63.4% were married, and 56.7% had one or more children.

Concerning descriptive statistics at the prefecture level, the proportions of respondents residing in prefectures with NPB teams, J1 League teams, and B1 League teams were 63.7%, 64.2%, and 55.6%, respectively. The average business climate index for each prefecture during the analysis period was 31.25 (a value above 50 means a “good” economy, and a value below 50 means a “bad” economy) [18], and the average number of cumulative COVID-19 cases (in hundreds) and deaths in the week leading up to the start of the survey wave was 4.0 and 5.0, respectively. These numbers translate to −0.7 and −0.8 when group-mean centered.

Among the respondents with the characteristics and contexts described above, the average spectator demand (i.e., the percentage of respondents who are willing to watch games of the relevant league) was greatest for NPB at 27.0%, followed by J.League at 13.8% and B.League at 5.8%. Table 1 describes the summary statistics of the respondents.

### 3.2. Trends of Spectator Demand

Table 2 illustrates the trends in the Japanese public’s game attendance demand in the three professional sports leagues and the infection status of COVID-19 from wave 1 to 8. The variation in the spectator demand for NPB, J.League, and B.League games during the pandemic was modest, with coefficients of variance (standard deviation/average × 100) of 5.3%, 7.0%, and 10.1%, respectively. There were no statistically significant alterations identified. In contrast, the COVID-19 data reflect the “second wave” from July to August (wave 3 and 4) and the “third wave” from November onward (wave 7 and 8), resulting in high volatility. The prefectural means of *Cases* and *Deaths* across each wave differed significantly at the 0.1% level (F statistics 2400.01 and 1013.78, respectively).

### 3.3. Factors Related with Spectator Demand

Table 3 shows the estimation results of the intercept-only model. The interpretation of these is straightforward, indicating what proportion of the total variation in spectator demand is attributable to each of the three levels. This is calculated as the ratio of the random prefecture variance (i.e., the intercept) to the total variance. For example, the prefecture-level variance of demand for NPB games attendance was 0.087, and the proportional variance was 2.58%. The calculation method was [0.087/(3.290 + 0.004 + 0.087)] × 100. Thus, approximately 2.58% of the variation in NPB spectator demand was due to differences among prefectures. There are at least two findings from Table 2. First, the variation in the Japanese public’s demand for professional sports games attendance is mainly due to individual factors, with more than 90% of the total variation located here. Second, spectator demand in COVID-19 crisis varies only very little between prefecture-waves.

Table 4 provides the estimation results of the multilevel model including the explanatory variables. Regarding the Level 1 variables, having an attendance demand for professional sports games was positively correlated with being a man, being younger, working full-time, being married, and having children. The largest spectator demand differences were observed between males and females (NPB: OR = 2.39, 95% CI = 2.22–2.56; J.League: OR = 2.56, 95% CI = 2.34–2.81; B.League: OR = 1.55, 95% CI = 1.36–1.77). The disparities between those who were employed full-time and non-full-time (NPB: OR = 1.30, 95% CI = 1.21–1.39, J.League: OR = 1.28, 95% CI = 1.17–1.40; B.League: OR = 1.55, 95% CI = 1.36–1.77), between married and unmarried people (NPB: OR = 1.25, 95% CI = 1.13–1.38; J.League: OR = 1.17, 95% CI = 1.03–1.33), and between those with children and without (NPB: OR = 1.17, 95% CI = 1.07–1.29; J.League: OR = 1.29, 95% CI = 1.14–1.46; B.League: OR = 1.42, 95% CI = 1.19–1.70) were comparable.

For the Level 3 variable, there is no surprise, but residents in prefectures with professional sports teams were more likely to have a stadium/arena attendance demand than those without teams. Disparities in spectator demand were found between NPB-prefectures and the others (OR = 1.67, 95% CI = 1.41–1.97), between B1 League prefectures and the others (OR = 1.50, 95% CI = 1.14–1.97), between J1 League prefectures and the others (OR = 1.30, 95% CI = 1.10–1.54), and between J2 League prefectures and the others (OR = 1.25, 95% CI = 1.06–1.49), indicating that, in the case of hierarchical leagues (i.e., J.League and B.League), division level is related to the spectator demand. Controlling for the presence of professional sports teams (i.e., Level 3 variables), the prefecture-level variation decreased from 0.06–0.15 (Model 1) to 0.04–0.11 (Model 2), resulting in a decrease in proportional variance from 1.74–4.30% to 1.16–3.13%.

Finally, none of the Level 2 variables were significantly related to spectator demand. To elaborate, during the study period of this research, there was no regular psychological response in terms of residents’ intention to attend professional sports games to the alterations in the number of COVID-19 cases or deaths within the prefecture. These results were robust to the changing COVID-19 case and death data as real numbers (i.e., no centering) or grand-mean-centered variables, or limiting the analysis sample to people living in prefectures with professional sports team (i.e., for NPB demand, *n_i_* = 12,827, *n_j_* = 88, *n_k_* = 11; for J.League demand, *n_i_* = 18,967, *n_j_* = 312, *n_k_* = 39; for B.League demand, *n_i_* = 16,815, *n_j_* = 240, *n_k_* = 30).

## 4. Discussion

There is mounting interest in the potential impact of public health crises on the behavior of individuals such as attendance at spectator sporting events [10,11,21,22]. Amid the global health emergency of the COVID-19 pandemic, a better understanding of individual behavioral/psychological responses is crucial for policymakers and sport business practitioners to control the spread of the virus and for risk management of professional sports clubs. In addition, although attendance demand in professional sport has long been discussed [23,24,25], most research, presumably because of data availability, has focused on North American and European regions, calling for more research in other countries such as those in Asia to assess the robustness of previously identified determinants of sport attendance demand [13].

In this study, we explore the relationship between local COVID-19 transmission and intention to attend games of major Japanese professional sports leagues (i.e., NPB, J.League, and B.League) utilizing data from a few repeated social surveys on sports spectator demand collected during the pandemic. We modeled attendance demand at multiple levels: (1) individual level (e.g., gender), (2) prefecture-wave level (e.g., local economic situation), and (3) prefecture level (e.g., the existence of quality teams in prefectures), with particular attention to the impact of newly confirmed cases and deaths of COVID-19, included in Level 2.

Reflecting on the descriptive results of spectator demand, during the COVID-19 crisis, 27.6–31.9% of the public had the intention to watch NPB games in stadiums, compared to 12.5–15.4% for the J.League and 5.2–7.2% for the B.League. Although not fully comparable due to different survey methods, pre-pandemic sport activity statistics show that demand was 26.3% for NPB, 11.5% for J.League, and 5.5% for B.League, which is relatively lower than the spectator demand data in this study [26]. This indicates that the spectator sports industry is unlikely to have lost “potential” customers due to the pandemic. This result echoes that of Reade et al. [11], who showed that sports fans are likely to return to the stands as soon as the stadium is reopened.

The current investigation revealed that COVID-19-related factors were not associated with intention-based spectator demand, which complements the interpretation of the related behavior-based spectator demand findings [9,10,11]. Gitter [9], for example, found that attendance demand for Mexican League Baseball significantly decreased during an epidemic event (i.e., H1N1). More in line with the current research context, Reade and colleagues [11] also demonstrated that the number of COVID-19 cases as well as deaths encouraged Belarusian football fans to stay at home during the first period of the pandemic, although their attendance demand quickly reverted. Combined with our results, it is likely that the reduction in the number of spectators due to the spread of the virus is due to risk perception and leisure constraints, rather than to a decrease in consumers’ preferences for sports spectating. There may also be more contextual reasons why spectator intentions were inelastic to COVID-19 factors. Japanese sport leagues and teams (NPB and J.League in particular) rapidly took measures to prevent the spread of COVID-19 in stadiums and arenas (e.g., spectator-less games, restricted admission) [27,28]. In fact, there have been no cases of spectators being infected by COVID-19, although a handful number of players and referees have tested positive. This might have contributed to public perceptions of safety in professional sport stadiums and arenas, resulting in the null effect of COVID-19 on spectator demand.

The findings indicated that attendance demand for professional sports games is explained by individual and prefecture-level team factors rather than COVID-19-related variables (i.e., confirmed active cases and deaths). Specifically, individuals who were male, younger, full-time employed, and had children reported higher attendance demand relative to their corresponding counterparts. Furthermore, the existence of professional sport teams in prefectures can lead to higher attendance demand than in prefectures without teams. This pattern of the results was prominent when a focal team belonged to a higher division within the league (i.e., higher team quality). The findings are consistent with the basic premise of the team quality principle, suggesting that higher-quality teams can provide more utility to fans, in turn, increasing their demand for sport [29]. Although the association between individual and prefecture-level team factors and attendance demand has been well documented [30,31], it is imperative to accumulate knowledge regarding the determinants of attendance demand in uncertain times. In this sense, the current research provided further evidence that individual factors and team quality still serve as influential antecedents of attendance demand even in the contexts of the COVID-19 epidemic.

There are several limitations to our study. First, the period of analysis is limited by the availability of spectator demand data. The SDS launched its repeated survey in May 2020, so data from the beginning of the outbreak (early 2020), when COVID-19 was an unknown infectious disease in Japan, are not included in our analysis. For the same reason, data from 2021 onward, when the infection spread further, are not included. Second, there are some missing variables. Some important variables in examining spectator demand, such as identification with favorite team, membership status, experience in attending sport events, distance to venue, and personal income, for example, were not included in the analysis, which was a major limitation of the secondary data analysis. Note that those Level 1 variables are theoretically uncorrelated with the COVID-19 variables, thus adding them to the regression model will certainly improve precision but will not change the estimates of the effects of *Cases* and *Deaths*.

## 5. Conclusions

In sum, data showed that (intention-based) spectator demand in Japanese major professional leagues is inelastic to public health emergencies. Rather, individual and prefecture-level team factors were found to be significant drivers. The findings can partly support the effectiveness of digital transformation. Given that spectator demand is maintained despite the leisure constraints imposed by the virus, practitioners in the sport industry can develop and diversify distribution channels by incorporating digital technologies. By doing so, unsatisfied demand due to the behavioral restriction can be remedied at least for a while. Turning to future research, it is recommended to observe the relationship between changes in the perception of risks and constraints and spectator behavior and intention during public health emergencies. In a similar sense, efforts should also be made to discover how the habituation effect on risk is generated.

## Figures and Tables

**Table 1 ijerph-19-05318-t001:** Definitions and descriptive statistics of variables used in the analysis.

Variables		Mean	SD	Min	Max
**Dependent variables**					
Spectator demand					
*NPB demand*	Nippon Professional Baseball attendance demand (1 = yes)	0.27	0.44	0	1
*J.League demand*	Japan Professional Football League attendance demand (1 = yes)	0.14	0.35	0	1
*B.League demand*	Japan Professional Basketball League attendance demand (1 = yes)	0.06	0.23	0	1
**Independent variables**					
**Level 1: Individual (*n* = 20,121)**					
*Gender*	1 = male, 0 = female	0.48	0.50	0	1
*Age*	Age (years)	50.07	16.87	18	94
*Employment*	1 = full-time, 0 = otherwise	0.45	0.50	0	1
*Marriage*	1 = married, 0 = otherwise	0.63	0.48	0	1
*Child*	1 = one or more, 0 = none	0.57	0.50	0	1
**Level 2: Prefecture-wave (*n* = 376)**					
*Economy*	Economic diffusion index reported by TDB Trends Research	31.25	4.00	20.80	42.80
*Cases*	Number of total COVID-19 cases in the week prior to the survey round/100 (prefecture-mean centered)	−0.66	4.52	−16.18	15.83
*Deaths*	Number of total COVID-19 deaths in the week prior to the survey round (prefecture-mean centered)	−0.80	7.96	−15.56	52.90
**Level 3: Prefecture (*n* = 47)**					
*NPB*	1 = prefecture with NPB team, 0 = otherwise	0.64	0.48	0	1
*J1 League*	1 = prefecture with J1 League team, 0 = otherwise	0.64	0.48	0	1
*J2 League*	1 = prefecture with J2 League team, 0 = otherwise	0.48	0.50	0	1
*J3 League*	1 = prefecture with J3 League team, 0 = otherwise	0.30	0.46	0	1
*B1 League*	1 = prefecture with B1 League team, 0 = otherwise	0.56	0.50	0	1
*B2 League*	1 = prefecture with B2 League team, 0 = otherwise	0.48	0.50	0	1

**Table 2 ijerph-19-05318-t002:** Changes in spectator demand and COVID-19 infection status between survey waves.

NPB Demand					J.League Demand				B.League Demand				COVID-19	
Yes			No		Yes		No		Yes		No		*Cases*	*Deaths*
n		%	n	%	n	%	n	%	n	%	n	%		
wave 1	466	31.0	1037	69.0	323	14.8	1857	85.2	111	5.7	1825	94.3	−4.64	2.01
wave 2	523	31.9	1115	68.1	374	15.4	2054	84.6	155	7.2	2007	92.8	−4.15	−3.59
wave 3	436	27.6	1143	72.4	293	12.5	2057	87.5	114	5.5	1976	94.6	−0.56	−5.12
wave 4	510	31.5	1109	68.5	318	13.3	2073	86.7	111	5.2	2009	94.8	0.46	−0.95
wave 5	474	29.1	1153	70.9	320	13.3	2079	86.7	124	5.8	2003	94.2	−1.85	−1.47
wave 6	463	28.6	1157	71.4	362	14.9	2061	85.1	137	6.4	1992	93.6	−1.73	−3.59
wave 7	464	28.6	1156	71.4	342	14.3	2047	85.7	126	6.0	1982	94.0	1.17	−2.09
wave 8	474	29.2	1147	70.8	337	14.0	2070	86.0	126	5.9	2017	94.1	5.63	8.51
chi^2^ (7)	13.189				13.443				9.681				—	—
F	—				—				—				2400.01 ***	1013.78 ***

*** denotes significant at the 0.1% level. The data on spectator demand are averages only for the prefectures with teams in each league. The values of cases and deaths are prefecture-mean centered, and cases are further divided by 100.

**Table 3 ijerph-19-05318-t003:** Results of multilevel logistic regression (intercept-only model) on spectator demand of Japanese citizens during the COVID-19 crisis.

	NPB Demand		J.League Demand		B.League Demand	
	OR	95% CI	OR	95% CI	OR	95% CI
Intercept	0.32 ***	[0.29, 0.35]	0.15 ***	[0.14, 0.17]	0.060 ***	[0.053, 0.069]
**Variance**						
Prefecture-waves	0.00		0.01		0.00	
Prefectures	0.09		0.06		0.15	
**VPC**						
Individuals	97.41%		98.26%		95.70%	
Prefecture-waves	0.12%		0.15%		0.00%	
Prefectures	2.58%		1.74%		4.30%	
*N* level 1	20,121		20,121		20,121	
*N* level 2	376		376		376	
*N* level 3	47		47		47	

*** denotes significance at the 0.1% levels.

**Table 4 ijerph-19-05318-t004:** Results of multilevel logistic regression on the relation between the three-level variables and spectator demand of Japanese citizens during the COVID-19 crisis.

	NPB Demand		J.League Demand		B.League Demand	
	OR	95% CI	OR	95% CI	OR	95% CI
Intercept	0.22 ***	[0.11, 0.42]	0.14 ***	[0.07, 0.32]	0.029 ***	[0.01, 0.09]
**Survey wave (ref: wave 1)**						
wave 2	1.11	[0.96, 1.28]	1.13	[0.94, 1.35]	1.11	[0.86, 1.45]
wave 3	0.89	[0.75, 1.06]	0.92	[0.74, 1.14]	0.82	[0.59, 1.14]
wave 4	1.03	[0.86, 1.25]	1.02	[0.81, 1.28]	0.80	[0.57, 1.13]
wave 5	0.99	[0.80, 1.23]	0.99	[0.76, 1.29]	0.78	[0.53, 1.15]
wave 6	0.98	[0.75, 1.26]	1.17	[0.86, 1.59]	0.81	[0.51, 1.27]
wave 7	1.00	[0.74, 1.35]	1.13	[0.79, 1.62]	0.71	[0.42, 1.20]
wave 8	0.93	[0.68, 1.28]	1.12	[0.77, 1.63]	0.74	[0.43, 1.29]
**Level 1: Individuals**						
*Gender*	2.39 ***	[2.22, 2.56]	2.56 ***	[2.34, 2.81]	1.55 ***	[1.36, 1.77]
*Age*	0.99 ***	[0.99, 0.99]	0.98 ***	[0.98, 0.99]	0.97 ***	[0.97, 0.98]
*Employment*	1.30 ***	[1.21, 1.39]	1.28 ***	[1.17, 1.40]	1.55 ***	[1.36, 1.77]
*Marriage*	1.25 ***	[1.13, 1.38]	1.17 *	[1.03, 1.33]	1.03	[0.86, 1.24]
*Child*	1.17 **	[1.07, 1.29]	1.29 ***	[1.14, 1.46]	1.42 ***	[1.19, 1.70]
**Level 2: Prefecture-waves**						
Economy	1.00	[0.97, 1.02]	0.99	[0.96, 1.02]	1.04	[1.00, 1.09]
Cases	1.00	[0.99, 1.01]	0.99	[0.98, 1.00]	0.99	[0.97, 1.01]
Deaths	1.00	[1.00, 1.01]	1.00	[1.00, 1.01]	1.00	[0.99, 1.01]
**Level 3: Prefectures**						
*NPB*	1.67 ***	[1.41, 1.97]				
*J1 League*			1.30 **	[1.10, 1.54]		
*J2 League*			1.25 *	[1.06, 1.49]		
*J3 League*			0.95	[0.79, 1.14]		
*B1 League*					1.50 **	[1.14, 1.97]
*B2 League*					1.14	[0.87, 1.51]
**Variance**						
Prefecture-waves	0.00		0.00		0.00	
Prefectures	0.05		0.04		0.11	
**VPC**						
Individuals	98.57%		98.84%		96.87%	
Prefectures	1.43%		1.16%		3.13%	
Prefecture-waves	0.00%		0.00%		0.00%	
*N* level 1	20,121		20,121		20,121	
*N* level 2	376		376		376	
*N* level 3	47		47		47	

*, **, and *** denote significance at the 5%, 1%, and 0.1% levels, respectively.

## Data Availability

The spectator demand data presented in this study are available to members of the Japan Society of Sports Industry.

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
