# Peer review of "COVID-19 and Attendance Demand for Professional Sport in Japan: A Multilevel Analysis of Repeated Cross-Sectional National Data during the Pandemic"

_ijerph, 2022, doi:10.3390/ijerph19095318_

Round 1

Reviewer 1 Report

Interesting paper that addresses the Attendance Demand for Professional Sport in Jappan. I applaud the authors for addressing this topic. In relation to the article, in my opinion it presents some aspects that make it difficult for the reader to understand, some paragraphs should be moved to other sections.

INTRODUCTION

The paper must end with the aim of the study, and during the introduction, paragraphs converge that are more typical of methodology, results or discussion. For example, from lines 63 to 74, it is clearly a paragraph that should be moved to the discussion.

In the last paragraph of the introduction, you must talk about the sections of the paper (method, results, discussion) and not sections.

METHODOLOGY

Has this study obtained any permission from an ethics committee? If so, they should indicate which committee approved it and what the code is. If according to Japanese law it is not necessary, they must also indicate it in the text.

Could authors define in this section how the Spectator Demand Survey (SDS) was developed? Note that it must be in this section and not in the Introduction.

One aspect that seems interesting to me is why the variable indoor vs. outdoor (eg I assume that basketball is indoor and football is outdoor). Perhaps, if the authors find it interesting, they can include it as a limitation of the study in the final section of the discussion.

RESULTS

correct in my opinion

DISCUSSION

Authors must include in the final part of the discussion the limitations of the study that you consider appropriate and may be relevant (for example, the absence of some variables that a posteriori would have been interesting)

CONCLUSION

Delete the following sentence, since it does not belong to the Line 322 conclusion (e.g., Pay-TV service, online streaming, virtual reality spectatorship)

Reviewer 2 Report

The Authors in this interesting study assessed the effect of COVID-19 on intention-based spectator demand for professional sports in Japan.

Line 67: January should have the year explained.

Table 1 is very difficult to read, dichotomous variables are usually reported as percentages of frequence and not absolute values.

Why was reported SD reported in a categorical value? In my opinion the Authors should rewrite the table for clarity. 

The "Economy" prefecture variable should be explained better: how was this index calculated?

COVID cases and deaths trends should be reported as percentages.

Line 179-181 needs a reference for the interpretation of the business index.

Why were cases and deaths divided by 100? in Table 2 the F line is not clear and should be explained better: does it mean that no result was staistically significant?

Why was a significance < 0.1% (< 0.001) chosen as value? This should be described in the Methods, standard significance is < 0.05.

Line 260  the reference number should be corrected

Line 279 the reference number should be corrected
